# Long-term variations in pH in coastal waters along the Korean Peninsula

Yong-Woo Lee[1], Mi-Ok Park[1], Seong-Gil Kim[1], Tae-Hoon Kim[2], Yong-Hwa Oh[3], Sang Heun Lee[4], DongJoo Joung[4,5]*

[1]Marine Environment Monitoring Department, Korea Marine Environment Management Corporation, Busan, Republic of Korea
[2]Department of Oceanography, Faculty of Earth Systems and Environmental Sciences, Chonnam National University, Gwangju, Republic of Korea
[3]Department of Convergence Study on the Ocean Science and Technology, Korea Maritime and Ocean University, Busan, Republic of Korea
[4]Department of Oceanography, Pusan National University, Busan, Republic of Korea
[5]Institute for Future Earth, Pusan National University, Busan, Republic of Korea

*Correspondence to*: DongJoo Joung (dongjoo.joung@pusan.ac.kr)

**Abstract.** Declining seawater pH, associated with rising atmospheric $CO_2$ levels, adversely affects marine organisms and ecosystems, thereby posing a considerable risk to coastal fisheries and economies. However, the effects of long-term pH variations in coastal waters remain poorly understood. In this study, we investigated the variability in pH in the coastal waters of Korea over an 11-year period (2010–2020) and sought to identify the principal drivers of pH fluctuations. Unlike the persistent pH declines observed in the open oceans and other coastal systems, Korean coastal waters showed no persistent pH variation, thus indicating that local biogeochemical processes may have a greater influence than atmospheric $CO_2$ in determining aquatic pH. Analysis of environmental data including temperature, salinity, chlorophyll *a*, and dissolved oxygen (DO) revealed a strong correlation between pH and DO. However, instances of pH changes exceeding those predicted by DO depletion alone indicate the influence of additional biogeochemical factors. As global seawaters warm, reduction in DO is anticipated to cause a further decline in the pH of coastal waters. This trend could have a pronounced influence on Korean coastal waters, which support extensive aquaculture operations integral to the local and national economies. Consequently, high-frequency monitoring is essential for extending the current time series and predicting future water quality.

**Graphical Abstract**

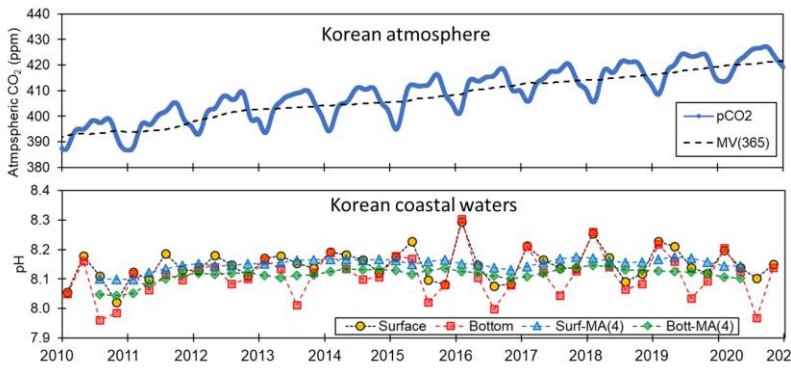


## 1 Introduction

The alarming increase in atmospheric carbon dioxide ($CO_2$) poses a significant threat to the global climate. Since the industrial revolution, atmospheric $CO_2$ concentrations have almost doubled, reaching 420 ppm in Mauna Loa, Hawaii

(https://gml.noaa.gov). Oceanic and terrestrial biospheres have been estimated to absorb approximately 50% of this increase in atmospheric $CO_2$ (Friedlingstein et al., 2022), leading to negative effects on marine organisms. Currently, it is estimated that oceans have absorbed approximately 26% of the total anthropogenic-derived $CO_2$ released between 1850 and 2020 (IPCC, 2022), with an estimated cumulative uptake of $170 \pm 35$ GtC, more than two-thirds of which have occurred post-1960. Compared with the $1.1 \pm 0.4$ GtC $yr^{-1}$ value obtained in the 1960s, the oceanic $CO_2$ sink increased to $2.8 \pm 0.4$ GtC $yr^{-1}$ during

the period from 2011 to 2020 (IPCC, 2022).

Although air–sea gas exchange is considered the primary pathway for oceanic $CO_2$ uptake, riverine inputs can make a significant contribution in the coastal areas of river-dominated estuaries (Cai, 2011). Having dissolved in seawater, $CO_2$ undergoes a series of chemical reactions, in which $CO_2$ reacts with water to form carbonic acid ($H_2CO_3$), which dissociates into hydrogen and bicarbonate ($HCO_3^-$) ions, of which the bicarbonate ions further dissociate to carbonate ions ($CO_3^{2-}$) by

releasing a further hydrogen ion. This process increases the concentration of hydrogen ions, thereby lowering the seawater pH and causing oceanic acidification (OA). Indeed, the increase in atmospheric $CO_2$ has already led to a reduction in oceanic pH, as evidenced by long-term datasets such as the Hawaii Ocean Time-series (HOT), the Bermuda Atlantic Time-Series (BATS), and the European Station for Time-series in the Ocean of the Canary Islands (ESTOC), which have recorded pH declines of approximately 0.02 units per decade since the 1980s (Solomon et al., 2007). Additionally, surface water pH in the open ocean

has dropped by 0.1 to 0.2 units since pre-industrial times due to anthropogenic-derived $CO_2$ (Pelejero et al., 2005), with projections indicating a further decline of 0.3 to 0.4 units by the end of the 21$^{st}$ century. This would correspond to an approximate 150% increase in $H^+$ concentrations ($[H^+]$) and a 50% reduction in $CO_3^{2-}$ concentrations (Orr et al., 2005).

Ocean acidification reduces carbonate ion concentrations in seawater by increasing $[H^+]$, posing a significant threat to calcareous marine organisms. The reduction in calcium carbonate ($CaCO_3$) saturation in surface waters, linked to OA, is a well-documented phenomenon that has been established based on the findings of models, field surveys, and long-term monitoring studies (Caldeira and Wickett, 2003; Feely et al., 2004 and 2008; Orr et al., 2005; Solomon et al., 2007). For example, long-term monitoring at the HOT station has revealed that surface water $pCO_2$ increases are correlated with atmospheric $CO_2$, driving increases in dissolved inorganic carbon and reductions in $CaCO_3$ saturation (Chen et al., 2023; Takahashi et al., 2009; Doney, 2008; Doney et al., 2007). Lower carbonate ion concentrations result in $CaCO_3$ undersaturation, thereby adversely affecting shell-forming marine organisms such as plankton, molluscs, echinoderms, and corals (Doney et al., 2007; Smith and Buddemeier, 1992; Kleypas and Yates, 2009; Feely et al., 2016; Orr et al., 2005; Kroeker et al., 2013; Nilsson et al., 2012). This undersaturation can reduce the growth, locomotion, reproductive capacity, and homeostasis in these organisms if they are unable to regulate calcification processes (Hendriks et al., 2015). Consequently, OA can have pronounced effects on the structure and function of marine ecosystems at multiple trophic levels.

In addition to the input of atmospheric $CO_2$, environmental changes can exacerbate OA. For example, along the Louisiana Shelf of the northern Gulf of Mexico, where severe bottom water hypoxia has been an annual feature since the mid-1980s, Cai et al. (2011) reported a pH reduction of 0.01 over the period between 1980 and 2010, which they attributed to anthropogenic atmospheric $CO_2$ inputs and $CO_2$ release during hypoxia. Similarly, these authors found evidence to indicate that pH declines in the East China Sea were linked to coastal eutrophication and low-oxygen-related $CO_2$ inputs. Globally, hypoxic regions are expanding in size and persisting for longer periods of time (Breitburg et al., 2018), thereby tending to indicate a future intensification of coastal pH declines.

Despite these general trends, the changes in oceanic $CO_2$ uptake and the resultant OA are not globally uniform. For example, the Mediterranean Sea has been established to absorb comparatively more atmospheric $CO_2$, leading to a reduction in pH of 0.0044 units per year (Hassoun et al., 2015; Palmieri et al., 2015; Flecha et al., 2015, 2019). In contrast, in the northern Pacific Ocean, the surface pH declined by only 0.0012 to 0.0003 units per year from 1965 to 2010, which is approximately four-fold lower than that observed in the Mediterranean Sea (Watanabe et al., 2018). This discrepancy in the rates of pH decline is assumed to be associated with regional variations in biogeochemical properties such as nutrient levels and freshwater inputs, which influence alkalinity (Flecha et al., 2015).

Although long-term pH monitoring is a well-established procedure in the study of open ocean systems, notably fewer studies have conducted in coastal zones. In contrast to the open oceans, in which pH tends to decline consistently in response to increasing atmospheric $CO_2$, coastal pH trends can differ significantly on account of their complex physical and biogeochemical dynamics, including nutrient and organic matter inputs from river runoff and groundwater discharge, oceanic forces (waves, tides, and currents), and human activities, all of which are subject to have seasonal variations, (Crossland et al., 2005, Flecha et al., 2012). Consequently, coastal pH values tend to vary widely and be characterized by different long-term trends (Borges and Gypens, 2010; Carstensen and Duarte, 2019; Bates et al., 2014). For example, Carstensen and Duarte

(2019) have reported a broad range of pH trends (-0.023 to 0.023 units yr$^{-1}$), predominantly associated with terrestrial inputs, whereas observations from Dutch coastal zones have revealed rates of pH change exceeding those predicted from $CO_2$ uptake alone, indicating significant influences from other biogeochemical processes, possibly associated with nutrient inputs (Provoost et al., 2010). However, given the multiple biogeochemical processes involved, coastal pH variations have not been well studied. For example, despite the presence of semi-closed bays, rivers, and fish farms that could influence aquatic pH changes, there have been no studies reporting long-term trend in pH in Korean coastal waters. The coastline of Southern Korea has ria landform with numerous semi-enclosed bays and fish farms. The western coast is characterized by high tidal prisms, extensive land reclamation, and large coastal cities, whereas the east coast has a narrow continental shelf, a geographically simpler coastline with no bays or fish farms, and is less populated, making it relatively undisturbed. Consequently, the diverse Korean coastline can serve as an excellent testbed for investigating how different biogeochemical settings affect pH over a long timescale. In this study, we accordingly investigated trends of pH, temperature, salinity, dissolved oxygen (DO), and chlorophyll *a* (chl *a*) in the surface and bottom coastal seawaters of Korea as part of ongoing monitoring programmes inaugurated in 2010. Using data collected from 2010 to 2020, we examined seasonal and spatial variations in pH and identified the predominant factors influencing these changes. Our results will provide a quantitative reference for future studies on coastal carbon dynamics and the impacts of atmospheric $CO_2$-induced warming and changing environmental conditions.

## 2 Methods and Materials

### 2.1 Data collection

The dataset used in this study was derived from a monitoring programme that has been conducted by the Korea Marine Environment Management Corporation (KOEM) since 2010 to monitor the quality of water along the entire Korean coastline (Fig. 1). This programme has included the monitoring of more than 279 sites at 64 locations, in which water quality parameters were measured in both surface and bottom waters four times annually (February, May, August, and November) (Supplementary Table 1). Detailed information on the programme can be found at KOEM (https://www.meis.go.kr). In this study, we examined data collected from 2010 to 2020, comprising approximately 28,966 measurements for each selected parameter, using which, we compiled two average datasets: one for the different coastal regions (south, west, and east), and one for the entire Korean coastline.

In this study, we analysed the long-term variations in seawater pH, temperature, salinity, DO concentrations, and chl *a* levels along the Korean coastline. All sampling locations were situated within 6 km of the shoreline, predominantly in shallow waters with depths < 10 m, except for a few sites along the eastern coast. These sites are influenced by multiple factors, including riverine inputs, land-use patterns, and climatic conditions, all of which are susceptible to the effects of global warming and climate change.

## 2.2 Site description

Korea's coastal geography is characterized by a highly mountainous east coast and plateau-like south and west coasts, giving rise to a less steep continental slopes in the south and west. Large river systems are found in the west and south, though not in the east. The west coast, bordering the Yellow Sea, features a typical drowned valley shoreline with shallow waters, a maximum depth of approximately 40 m, and extensive muddy tidal flats, which account for approximately 3% of the world's entire tidal flats. This area experiences high water turbidity, macrotidal conditions (tidal amplitudes up to 5 m), and has undergone substantial land reclamation for agricultural, industrial, and residential uses, which have contributed to altering currents, sediment transport, and biogeochemical properties (Williams et al., 2014, 2015). The coastline is influenced by the Korean Coastal Current, which receives inputs from the Kuroshio Warm Current in summer and the Yellow Sea Cold Current in winter (Hwang et al., 2014), and three major rivers (the Han, Geum, and Yeongsan), along with numerous streams, drain into this region.

The southern coast features ria estuaries with moderate relief and V-shaped valleys. It is dotted with numerous islands and semi-enclosed bays, resulting in slow currents, and a semi-diurnal tidal range of 1 to 3 m, and also supports extensive fish farming operations (Williams et al., 2013). Major rivers such as the Nakdong and Seomjin, along with numerous tributaries, discharge freshwater into the coastal waters. Recent dam construction in the Nakdong River system has, however, significantly reduced freshwater and nutrient inputs in this region (Williams et al., 2013). The southern seas are influenced by the Tsushima Current, a branch of the Kuroshio Current, which transports high-temperature, high-salinity waters with occasional inputs from the Changjiang diluted water during summer (Lie and Cho, 1994). Compared with the southern and western coasts, the eastern coastline of Korea is relatively simple and monotonous. Due to its mountainous terrain, the eastern coast features a narrow continental shelf with a steep slope, and given that there are no major rivers with outlets along this coast, most materials, including nutrients and organic matter, are transported via the Tsushima Current, which flows along the southern coast of Korea (Jang et al., 2013).

## 2.3 Water collection and measurement

To examine the temporal and spatial variations in marine environmental parameters (temperature, salinity, pH, DO, and chl *a*), seawater samples were collected using 5-L Niskin bottles attached to a rosette sampler deployed from onboard the KOEM research vessel. Seawater temperature and salinity were measured using a CTD profiler (Seabird 19plus; Sea-Bird Electronics Inc., USA). For pH measurements, we used a portable sensor (Orion Star A329; Thermo Scientific, USA) with an accuracy of ±0.002 pH units. To minimize temperature changes, all pH measurements were performed onboard immediately after water collection. Briefly, water samples were collected from both surface and bottom depths (1 m from the seafloor) using Niskin samplers, and upon retrieval, the water samples were transferred to glass containers (a beaker) using a silicone tube. Immediately after transfer, the pH sensor was immersed in the water sample and pH readings were recorded having initially ensured that the values had stabilized, typically within a few tens of seconds. Calibration of the pH sensor was

conducted daily prior to sample measurements using three standard buffer solutions (pH 4, 7, and 10) in accordance with the

manufacturer's guidelines. The DO in seawater was measured using the Winkler-Sodium Azide titration method. For chl *a* analysis, seawater samples (1 L) were collected from the surface and bottom layers and filtered onboard using 0.45-μm pore-sized membrane filters (47 mm diameter), which were then stored at -20 °C until further analysis. Chl *a* was extracted from the filters and measured using a fluorometer (10-AU model; Turner Designs, USA) in the laboratory.

## 2.4 Data processing

The descriptive statistics for the complete dataset are presented in Supplementary Tables 1–3. For the period 2010–2020, each parameter included over 14,000 data points from 64 different locations and 279–355 sites, which were averaged for each sample collection period to produce 44 data points per parameter over the study period. These averages were calculated for the entire study area, and for three distinct coastal regions (south, west, and east) to investigate inter-annual trends over the study period. Additionally, the data were classified into three geographical categories, namely, estuaries, bays, and river

mouths, data for which were collected from 47, 14, and 3 locations, respectively. On the basis of the locations, we initially specified the river mouth locations, the Nakdong, Youngsan, and Guem rivers, corresponding to sites on the Nakdong, Mokpo, and Gunsan coastlines, respectively. Thereafter, we identified 14 locations in the bay system, and the remaining locations were classified as estuaries (Supplementary Table 1). Temporal trends throughout the study period were further analysed using a centred moving average technique to remove seasonal fluctuations, while hierarchical cluster analysis was employed to identify

the closest link to pH. Both analyses were performed using IBM SPSS Statistics 26 (USA).

We also obtained estimates for $CO_2$-derived pH variations, for which $CO_2$ production was inferred from apparent oxygen utilization (AOU) using the Redfield ratio (106/138 for $CO_2/O_2$, Cai et al., 2011). The AOU was calculated by subtracting the measured dissolved oxygen concentration from the saturation concentration derived from temperature and salinity (i.e., $AOU = DO_{saturated} - DO_{measured}$), and thus, positive values of AOU indicate a net consumption (or depletion) of

DO. To represent the degree of DO depletion or repletion, AOU was expressed as a percentage of DO depletion (or consumption) by calculating the ratio of AOU to $DO_{saturated}$. pH variations were subsequently calculated using CO2SYS software, based on the following conditions: dissolved inorganic carbon (1986 μmol/kg), total alkalinity (2228 μmol/kg) (Hwang and Lee, 2022), total dissolved phosphate (1 μM), and total dissolved silicate (15 μM) concentrations (KOEM). In this regard, it is important to note that the estimations assume that all $CO_2$ produced from oxygen depletion has a direct

influence on pH, which may not invariably be the case, particularly in shallow coastal systems, wherein the $CO_2$ in water can be released into the atmosphere depending on currents, winds, and tides. Therefore, our assumption may overestimate the extent of pH decrease.

# 3 Results and Discussion

## 3.1 Long-term trends in pH

Given its profound impact on aquatic biota, OA is of considerable concern for global warming (Raven et al., 2020; Hinga, 2002). Consequently, gaining an understanding of the OA in Korean coastal waters, in which a considerable number of fish and seaweed farms are sited, is of particular importance. Variations in the average pH levels in all Korean coastal waters are depicted in Figures 2–5 (Supplementary Figures 1–6 provide a comprehensive dataset). Monthly averages for February, May, August, and November of each year revealed a pH range of between 7.96 and 8.30 (mean, 8.13; n = 88). Notably, the

pH values recorded for bottom waters were found to be slightly lower than those of surface waters, which we assume to be attributable to the decomposition of organic matter releasing $CO_2$ in bottom waters, biological production, and the associated DO production in surface waters. Over the 11-year timeframe, monthly averaged pH levels displayed a discernible increasing trend in both surface and bottom waters, with respective slopes of 0.000029 $yr^{-1}$ (± 0.000007) and 0.000035 $yr^{-1}$ (±0.000009) (Supplementary Table 4). Although this regression trend appeared to be statistically significant, with an F-value of 0.0003,

these low regression slopes indicate that Korean coastal waters have not undergone any pronounced reduction in pH at least during the study period. This observed pH trend contrasts somewhat with the patterns reported in previous studies, which have observed declining pH trends in both coastal and open ocean systems. For example, reductions in pH have been documented in other coastal regions worldwide, such as the Mediterranean Sea (Flech et al., 2015), Dutch coasts (Provoost et al., 2010; Hofmann et al., 2011), the Washington coast, USA (Lowe et al., 2019), and a number of other areas (refer to review articles

by Duarte et al., 2013; Carstensen and Duarte, 2019; Wootton et al., 2008). Notably, these studies have reported substantial rates of decline, ranging from -0.045 to -0.0044 $yr^{-1}$. Moreover, Carstensen and Duarte (2019) observed that at certain sites (21 of 83), the pH of coastal waters declined even more rapidly than the rate of decline observed in the open ocean (-0.0018 $yr^{-1}$), driven by rising atmospheric $CO_2$ and variations in local salinity and primary production.

     In regions geographically proximal to Korea, coastal waters in Japan and China have shown a decline in pH levels.
For example, Ishida et al. (2021), who conducted a 30-year monitoring study from 1980 to 2010 at two coastal sites in Japan, revealed trends in pH change ranging from 0.0032 to -0.0068 $yr^{-1}$, with more pronounced declines being observed along the Pacific coastline than along the East Sea (or Sea of Japan) coast. Ishida et al. (2021) and Ishizu et al. (2019) have suggested that such regional differences could be attributed to local biophysical and chemical factors, with the Pacific coastline being influenced by the meandering Kuroshio Current, which is mainly governed by atmospheric $CO_2$ absorption and coastal

eutrophication, leading to low DO concentrations in bottom waters, and vertical mixing off the East Sea coast. Recently, Ishizu et al. (2019) have reported long-term pH trends from over 280 sites in Japanese coastal waters based on monitoring data collected over a period of more than 30 years, which have revealed that in a majority of the areas assessed (>75%), the annual maximum pH exhibited a decline of -0.0024 $yr^{-1}$. Moreover, Cai et al. (2011) found that regions such as the East China Sea and Louisiana Shelf, which are characterized by seasonal bottom-water hypoxia, experienced more severe pH declines than

could be attributed solely to atmospheric $CO_2$. However, the findings of all these previous studies indicate that, for coastal waters, variations in pH could be governed predominantly by local biogeochemical properties.

Conversely, some sites have been found to have undergone an increase in pH, as revealed in the aforementioned study by Ishizu et al. (2019), in which the remaining 25% of assessed sites showed an increasing trend, although these authors were unable to discern any evident geographical patterns among the sites. Additionally, Carstensen and Duarte (2019) documented
a wide range of pH trends, with slopes ranging from -0.023 to 0.023 $yr^{-1}$. Their analysis revealed that though a larger proportion (46 of 83 sites) of coastal areas experienced a pH decline, the remaining 37 sites displayed either increasing trends or negligible changes, mirroring our observations in the present study. Hence, despite the overarching influence of global atmospheric $CO_2$ forcing, there are substantial regional variations in pH fluctuations. Consequently, the lack of any marked changes in pH along the Korean coastline is far from unique and reflects the complexities of pH dynamics in coastal waters, wherein local
biogeochemical properties (e.g. biological productivity, chemical inputs from rivers and sediments, and alkalinity) differ from those in the open ocean and among different regions.

## 3.2 Local pH variations

Although we established that overall, the variations in pH along the entire Korean coastline were not substantial, the localized changes in specific coastal regions exhibited fine-scale seasonal and vertical dynamics. Generally, pH levels were
higher along the southern and eastern coasts (~8.15) than along the western coast (~8.10). Moreover, seasonally, peaks and troughs in pH were observed in February and August along the southern and western coasts, respectively, whereas compared with the other regions, we observed no consistent patterns in seasonality on the east coast (Figs 3-5). Furthermore, while we failed to detect any appreciable disparities in the pH of surface and bottom waters along the western coast, notable differences were observed between the surface and bottom waters along the eastern and southern coasts, particularly in May and August
(Figs 3-5).

Spatial variations in pH are assumed to be influenced by physiographical features. The western coast of Korea is characterized by shallow nearshore waters and a notably high tidal prism (up to 5 m). Given that our sampling sites along the western coastline predominantly had depths of less than 10 m, the substantial tidal prism and semi-diurnal tidal cycle would be presumed to promote homogeneous vertical mixing. Conversely, the eastern coast features a narrow continental shelf, with
water depths exceeding 100 m just a few kilometres offshore. Unlike the western coast, the tidal prism along the eastern coast is approximately 1 m, and coupled with seasonal variations in coastal currents (Chang et al., 2000), this contributes to a persistent stratification, which was evident in nearly all assessed water quality parameters, with the only exception being the patterns observed in February (Fig. 5).

In terms of these physiographical characteristics, the southern coastline can be considered to be characterized by
conditions intermediate between those found in the other two regions, with a tidal prism of 3 m and an average water depth of a few tens of meters. However, the southern coastline is characterized by numerous semi-enclosed bays with limited water

exchange and the frequent occurrence of bottom-water hypoxia (Lee et al., 2018; Lee et al., 2021; Huang and An, 2022), which contribute to the development of more pronounced vertical gradients in water quality parameters, including pH, during summer compared with winter (Fig. 3). For example, bottom-water hypoxia induces the release of $CO_2$ and subsequent accumulation in the stratified water column, and eventually, this $CO_2$ will cause a reduction pH (Cai et al., 2011).

Among the assessed aquatic systems, we found that variation in the average pH within estuaries was relatively narrow, ranging from 8.00 to 8.25, whereas that in the other two systems showed a notably wider range, from 7.75 to 8.4. Moreover, compared with the estuarine and river mouth sites, we detected more pronounced surface-to-bottom differences in average pH in the bay systems, with the bottom-water pH in bays being generally lower than that in the other locations. This difference can probably be ascribed to the lower DO concentrations in the bottom waters of bays, thereby contributing to $CO_2$-induced acidification. The inter-annual trends in pH and other parameters are presented in Supplementary Table 4. Notably, within all three of the assessed coastal regions, temperature showed a clear increasing trend of 0.001°C per year in both surface and bottom waters, whereas the other assessed parameters, including pH, showed no significant trends. Overall, the three coastal regions (south, west, and east) and three assessed aquatic systems (estuary, bay, and river mouth) showed no significant inter-annual trends with respect pH or other parameters across all regions.

### 3.3 Factors influencing pH variations

The variations in pH within coastal systems are influenced by a multitude of physical, biogeochemical, and ecological factors and processes, including temperature, salinity, DO, and biological production/respiration, which influence gas solubility, alkalinity, and $CO_2$ dynamics within the water column (Doney, 2008; Dore et al., 2009; Duarte et al., 2013). Additionally, by affecting open ocean endmember conditions, fluctuations in currents can influence pH values (Ishida et al., 2021).

By modulating gas solubility, water temperature can play a key role in pH regulation, with colder water having the capacity to retain larger amounts of gases than warmer water, whereas warmer water promotes degassing, which often results in lower pH levels. Additionally, increases in temperatures can alter thermodynamic properties, raising the dissociation constants of carbonic acid ($H_2CO_3$) and bicarbonate ($HCO_3^-$), thereby increasing acidity. However, the most pronounced effects of temperature on pH are associated with its influence on different biogeochemical processes. For example, elevated temperatures can intensify water column stratification, thereby contributing to bottom-water hypoxia, which leads to $CO_2$ accumulation and redox reactions that further promote reductions in pH. However, in coastal environments, the effects of temperature on pH are typically less pronounced than those attributable to other factors such as DO concentration and primary productivity. Consequently, long-term pH trends may show a negative correlation with temperature. In this study, we have not observed significant correlation between pH and temperature in either the surface or bottom waters (Supplementary Figs 7-10 and Supplementary Table 5). Despite a clear increasing trend in water temperature, at a rate of 0.001 yr$^{-1}$, observed across all

regions of the Korean coast (Fig. 2 and Supplementary Tables 4 and 5), the weak correlation between pH and temperature
suggests that temperature is not the primary driver of pH variation in Korean coastal waters.

    Salinity serves as an indicator of freshwater input, which typically has a lower alkalinity and is a key factor influencing pH dynamics (Saraswat et al., 2015; Carstensen and Duarte, 2019). However, watersheds underlain by limestone bedrock typically exhibit higher alkalinity than oceanic endmembers. Although this not a characteristic of the watersheds in Korea, in which alkalinity generally remains within the typical range for rivers (Hwang and Lee, 2022), a positive correlation between
pH and salinity is often observed in regions with low salinity (<20), in which the buffering capacity is minimal (Carstensen and Duarte, 2019). At most of our study sites, salinity was greater than 23, which would tend to indicate a sufficient alkalinity to buffer any moderate changes pH. Nevertheless, we failed to observe any correlations between pH and salinity (Supplementary Table 5), implying that salinity (or freshwater input) may not be the primary driver of pH changes in Korean coastal waters.

Primary production, as indicated by chl $a$ levels, can directly influence pH by consuming $CO_2$ via photosynthesis, particularly in surface waters (Soetaert et al., 2007). Accordingly, elevated levels of pH can be detected during the hours of daylight as a consequence of photosynthetic activity (Cummings et al., 2019; Wootton et al., 2008; Ono et al., 2023). However, over the 11-year period of the present study, we detected considerable differences between the variations of chl $a$ and pH (Supplementary Table 5), which again may indicate that primary production may not be the main driver of the observed
variation in pH. However, it is noteworthy that the relationship between chl $a$ and pH has primarily been documented in controlled settings (e.g. closed chamber experiments; Cummings et al., 2019) or over short time scales (e.g. daily observations; Wootton et al., 2008; Ono et al., 2023). This relationship tends to be less evident in natural environments, which could be associated with the time lag or varying response times of biological productivity (chl $a$) to environmental changes (e.g. pH). Thus, despite our observation of an apparent decoupling between pH and chl $a$ in the present study, pH can still be significantly
affected by primary production processes (Duarte et al., 2013; Ono et al., 2023).

    Reduced DO concentrations are primarily driven by biological respiration within the water column and sediments, which is associated with the release of $CO_2$. This liberated $CO_2$ can accumulate in the water column, particularly in the presence of the strong density gradients that are typically observed during summer. Once released, $CO_2$ reacts with water molecules, producing hydrogen ions and subsequently lowering the pH (Cai et al., 2011; Lowe et al., 2019). Thus, pH can exhibit a
positive correlation with DO, a relationship that we observed in both surface and bottom waters (Supplementary Table 5). In this regard, previous studies, such as that conducted by Cai et al. (2011), have reported pronounced declines in pH under conditions of low DO (<64 μM) along the Louisiana Shelf, wherein pH levels fell below 7.8. However, given that the monthly average DO concentrations along each of the Korean coastline never dropped below 128 μM, the pH levels observed in the present study were not as low as those measured on the Louisiana Shelf, but were still notably lower, approximately 7.93,
compared with normal conditions in Korean waters (8.20). To gain a better understanding of the DO consumption and pH changes, we calculated the apparent oxygen utilization (AOU), of which the positive values indicate a net DO consumption (or depletion). Among all the assessed regions and seasons, we failed to detect any significant consumption of DO in surface

waters (Figs 2-5), whereas contrastingly, in the bottom waters, pH showed a strong negative correlation with DO depletion (Fig. 6), indicating that an influx of $CO_2$ within the water column promoted a reduction in pH. For example, along the south coast, values of AOU reached up to 52.2 μM (~23.6%), which equates to 40.1 μM of $CO_2$ when converted using the Redfield ratio (106/138 for $CO_2/O_2$; Cai et al., 2011). This influx of $CO_2$ could theoretically lead to a reduction in pH of approximately 0.08 pH units. Moreover, the change in pH induced by the released $CO_2$ resulting from DO consumption was within the range of the observed pH variation (i.e. a 0.54 difference between 8.40 (maximum) and 7.86 (minimum)) (Fig. 7). Furthermore, the pH estimated from the AOU derived $CO_2$ exhibited a strong correlation with the measured pH (Fig. 7). These findings thus provided evidence to indicate that the pH in Korean coastal waters is primarily controlled by oxygen conditions and associated $CO_2$ dynamics, which is similar to observations in the Louisiana Shelf and East China seas, wherein severe hypoxia occurs annually (Cai et al., 2011).

The findings of cluster analysis perform in this study revealed consistent hierarchical patterns among all coastal regions and within the three areas (Fig. 8 and Supplementary Fig. 10). Specifically, pH and DO concentrations were consistently linked in the first cluster among the study sites, which is consistent with the findings of our linear regression analysis, thereby providing further support for the significant role of DO in influencing pH variability. However, the secondary links revealed certain variations. On the south coast, chl *a* concentration was identified as the second link for both surface and bottom waters, as well as for surface waters from the west coast. In contrast, salinity was found to be the second link for both the surface and bottom waters on the east coast, and for bottom waters along the west coast. Among the three types of aquatic system assessed, salinity was the primary factor influencing pH in estuarine surface waters, whereas in bays and river mouths, DO was identified as the dominant factor. We speculate that these regional differences in secondary linkages could be attributed to variations in biological productivity, in which areas with higher primary production, such as the southern and western coasts, appear to be characterized by relatively stronger pH modulation due to primary productivity. Conversely, in regions such as the east coast in which primary production is lower, salinity may play a more prominent role. Nevertheless, the influence of salinity on pH along the east coastline remains unclear, as this region was found to be characterized a minimal variation in salinity. Despite these uncertainties, the results of cluster analysis consistently highlighted the sensitivity of pH to DO conditions, thereby tending to indicate that DO has a more substantial impact on pH than other environmental parameters.

**3.4 pH variation in Korean coastal waters: insights and implications**

The 11-year pH monitoring dataset obtained in this study provides an invaluable resource for examining the pH conditions in coastal waters with respect to rising atmospheric $CO_2$ levels. Notably, during this period, there has been a considerable increase in atmospheric $CO_2$ concentrations along the Korean coastline, which is possibly one of the highest rates globally (Fig. 9). Between the years 2010 and 2020, atmospheric $CO_2$ concentrations rose from 395 to 424 ppm, at a rate of 2.64 ppm per year on Anmyeon Island, Korea, which is greater than the rate of 2.27 ppm per year recorded at Mauna Loa, Hawaii (https://gml.noaa.gov), which results in a pH decrease of approximately 0.002 per year (Solomon et al., 2007).

Nevertheless, despite this prominent increase, coastal pH variations may not be strongly influenced by atmospheric $CO_2$ alone, given the potential additional contribution of factors such as temperature, salinity, DO, primary production, hydrological processes, land use, nutrient inputs, pollution, and river water inputs. Conversely, pH levels in open oceans, which are less influenced by these factors, may be affected to a greater extent by increases in atmospheric $CO_2$.

345        In this regard, the East (Japan) Sea (EJS) offers a prime opportunity to compare changes in the pH of coastal and open-ocean environments. As a deep marginal sea with depths exceeding 3,000 m, the EJS serves as a net sink for atmospheric $CO_2$, absorbing approximately 1% of anthropogenic $CO_2$ emissions (Kim et al., 2022; Kim et al., 2023). Long-term pH monitoring in the central EJS from 1965 to 2015 has revealed a declining trend in pH at a rate of -0.0016 $yr^{-1}$ (Chen et al., 2017). Similar trends have been observed in the North Atlantic (1995–2003) and at the BATS (1983–2005), with pH reductions

of -0.0017 $\pm$ 0.0005 $yr^{-1}$ and -0.0017 $\pm$ 0.0001 $yr^{-1}$, respectively (Gonzalez-Davila et al., 2007; Bates, 2007). Consistently, pH reduction of -0.0019 $\pm$ 0.0002 $yr^{-1}$ has been reported in the surface waters of the Pacific Ocean, based on in situ measurements from the HOT from 1988 to 2007 (Dore et al., 2009). These findings indicate that the reductions in the pH in the central EJS mirror the trends in other open ocean regions, and thus suggest that the EJS shows carbonate system characteristics typical of those of the open ocean. Consequently, it is reasonable to assume that the observed decline in pH can

probably be attributable to a gradual increase in atmospheric $CO_2$ over time.

        Coastal regions bordering the central EJS are characterized by distinct pH variations relative to the open ocean and other coastal areas of the EJS. Whereas we found that the eastern coast of Korea, west of the EJS, showed no significant reduction in pH over the course of the study period, the Japanese coast, southeast of the EJS, has been found to show a pronounced pH decline, with rates surpassing those of the central EJS. Studies by Ishizu et al. (2019) and Ishida et al. (2021)

have reported acidification rates of -0.0025 and -0.003 $yr^{-1}$, respectively, based on pH measurements from 1978 to 2009 at Niigata and over a 30-year period at Kashiwazaki Bay. Although comprehensive datasets that could be used to identify the precise causes of these spatial variations are lacking, local biogeochemical factors, such as nutrient levels, biological production, oxygen conditions, and contaminant inputs, are assumed to play significant roles. For example, compared with the Japanese coast, primary production is notably higher along the Korean coast (Joo et al., 2015). In addition, higher amounts of

contaminant inputs including heavy metals reducing primary production due to biotoxicity, or eutrophication causing bottom water hypoxia in the vicinity of Niigata, a megacity on the Japanese coast, may have contributed to these observed differences. Nonetheless, the rate of pH decline in Japanese coastal waters has been more rapid than that recorded in the central EJS (-0.0016 $yr^{-1}$) and Korean coastal waters. However, the factors contributing to this accelerated acidification in Japanese coastal waters have yet to be sufficiently determined (Ishida et al., 2021). Overall, the observed variations in pH indicate strong spatial

heterogeneity governed by complex local biogeochemical conditions in coastal waters, which contrasts with the more uniform pH trends observed in open ocean systems.

Global ocean warming is anticipated to exacerbate the ongoing decline in seawater pH, as a consequence of elevated degassing. Along the Korean Peninsula, seawater temperatures have shown an increasing trend in both surface and bottom waters over the past few decades. This warming could accelerate the decline in pH along the Korean coast, potentially outweighing the influence of local heterogeneity (Breitburg et al., 2018). This decline in the pH of Korean coastal waters is particularly concerning given its potentially detrimental impacts on diverse aquacultural facilities, including those culturing shellfish, fish, and seaweed, the stocks of which generally have species-specific tolerance ranges (Kroeker et al., 2013; Lowe et al., 2019). Thus, oceanic acidification can have serious economic implications for both the local and national economies.

**4 Conclusion**

The dataset compiled in this study enabled us to examine pH variability in shallow coastal waters along the Korean coast over a 11-year period. Despite a general trend of pH decline recorded elsewhere in both open and coastal waters due to rising atmospheric $CO_2$, our data failed to provide any clear evidence of a pH decline in Korean coastal waters, thereby indicating that increasing surface water $CO_2$ levels are not the primary drivers of pH variations observed in these waters. Instead, we provide evidence to indicate that pH variations are closely linked with DO levels in almost all coastal regions (i.e. south, west, and east) and within the three assessed geographical areas (estuary, bay, and river mouth). This putative relationship between pH and oxygen levels implies that global warming, which reduces DO by degassing and respiration in response to a strong thermal gradient, could lead to a decline in pH in Korean coastal waters. Such a reduction in pH could have significant implications given that Korean coastal waters support numerous fish and seaweed farms raising stock that are particularly sensitive to the chemical properties of water. Nonetheless, our findings provide a baseline for further studies examining variations in the pH of the shallow coastal waters of Korea. Overall, future studies and monitoring efforts are essential to enable reasonable predictions of the impacts of ocean acidification on marine ecosystems and the bioeconomy of Korea's coastal waters.

**Data Availability**: The datasets analysed in this study can be found in the National Marine Environmental Measuring Network programme. The data are available at www.meis.go.kr.

**Author Contributions**: YWL, MOP, and SGK contributed to the data acquisition and sample collection. DJJ processed the data. YWL, MOP, SGK, THK, YHO, SHL, and DJJ interpreted the biogeochemical data. DJJ wrote the first draft of the manuscript. All authors contributed equally to the review and editing of this manuscript and have approved the submitted version.

**Competing Interests:** The authors declare that they have no conflicts of interest.

**Supplementary Materials:** The Supplementary Material for this article can be found online.


**Acknowledgements**

The authors are grateful to the crew members of the R/V from KOEM for their assistance. Constructive comments from the anonymous reviewers have enabled us to substantially improve the manuscript. This research was supported by the Global-Learning & Academic Research Institute for master's and PhD students and a Postdoctoral Program (G-LAMP) of the National

Research Foundation of Korea (NRF) grant funded by the Ministry of Education (No. RS-2023-00301938) and a National Research Foundation of Korea Grant funded by the Korean Government (NRF-202215000003).

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

Figure Captions:

Fig. 1. Sample collection sites along the Korean peninsula. The base map was downloaded from
https://server.arcgisonline.com.

Fig. 2. Monthly and centred moving averages of the parameters for the entire Korean coasts. (a) temperature, (b) salinity, (c) pH, (d) dissolved oxygen (DO) depletion, and (e) chlorophyll *a* concentration for the entire seas along the Korean coasts during the 2010–2020 period.

Fig. 3. Monthly and centred moving averages of the parameters on southern coast of Korea. (a) temperature, (b) salinity, (c)
pH, (d) dissolved oxygen (DO) depletion, and (e) chlorophyll *a* concentration for the south coasts of Korea during the 2010–2020 period.

Fig. 4. Monthly and centred moving averages of the parameters on western coast of Korea. (a) temperature, (b) salinity, (c) pH, (d) dissolved oxygen (DO) depletion, and (e) chlorophyll *a* concentration for the west coasts of Korea during the 2010–2020 period.

Fig. 5. Monthly and centred moving averages of the parameters on eastern coast of Korea. (a) temperature, (b) salinity, (c) pH, (d) dissolved oxygen (DO) depletion, and (e) chlorophyll *a* concentration for the east coasts of Korea during the 2010–2020 period.

Fig. 6. Relationships between pH and dissolved oxygen (DO) depletion (%). (a) the entire Korean coast, and (b) the south, (c) west, and (d) east coasts. DO depletion was calculated by subtracting the measured concentration from the temperature-salinity

derived saturation (i.e., DO depletion (%) = $(DO_{saturation} - DO_{measured})/DO_{saturation} * 100$). Thus, a positive value of DO depletion indicates oxygen consumption.

Fig. 7. Distribution of measured and estimated pH. pH estimation was calculated at conditions of 35, 25°C, 1 µmol/kg, 15 µmol/kg, 2228 µmol/kg, and 1986 µmol/kg (initial) salinity, temperature, phosphate, silicate, total alkalinity, and total $CO_2$ concentrations, respectively. With the exception of $CO_2$ concentrations, other parameters were not altered for the estimation.

$CO_2$ concentrations were altered based on dissolved oxygen (DO) consumption and the Redfield ratio (106/138 for $CO_2/O_2$).

Fig. 8. Hierarchical cluster analysis of parameters in coastal waters for: a) and b) the entire, c) and d) the south, e) and f) the west, and g) and h) the east coasts of Korea. The left columns (a, c, e, g) represent surface waters, while the right (b, d, f, h) columns represent bottom waters.

Fig. 9. Trends of atmospheric $CO_2$ and seawater pH. Variations in (a) atmospheric $CO_2$ (ppm) at Anmyeon Island and (b)

seawater pH along the entire Korean coast.

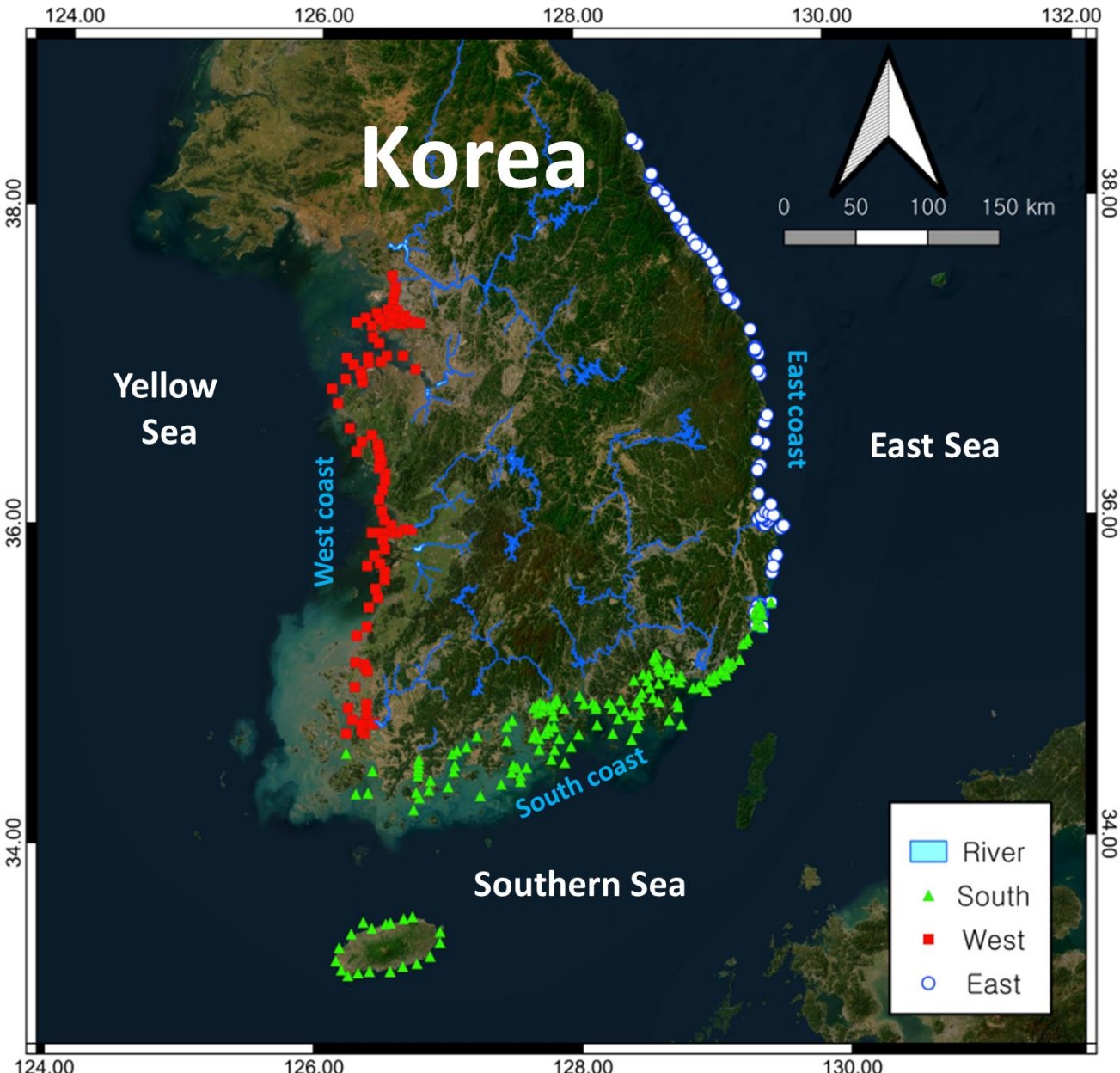

Figure 1.

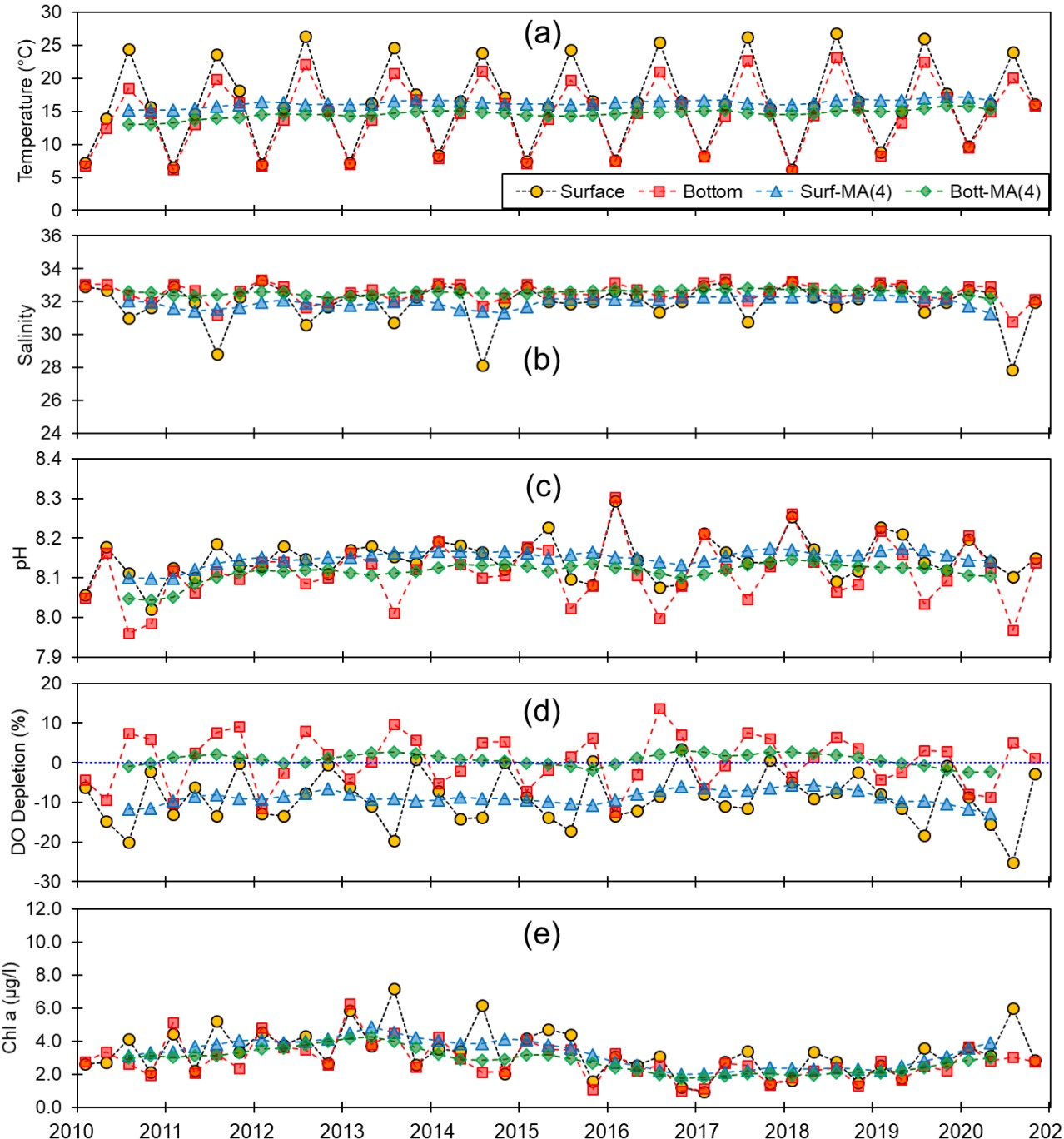

Figure 2.

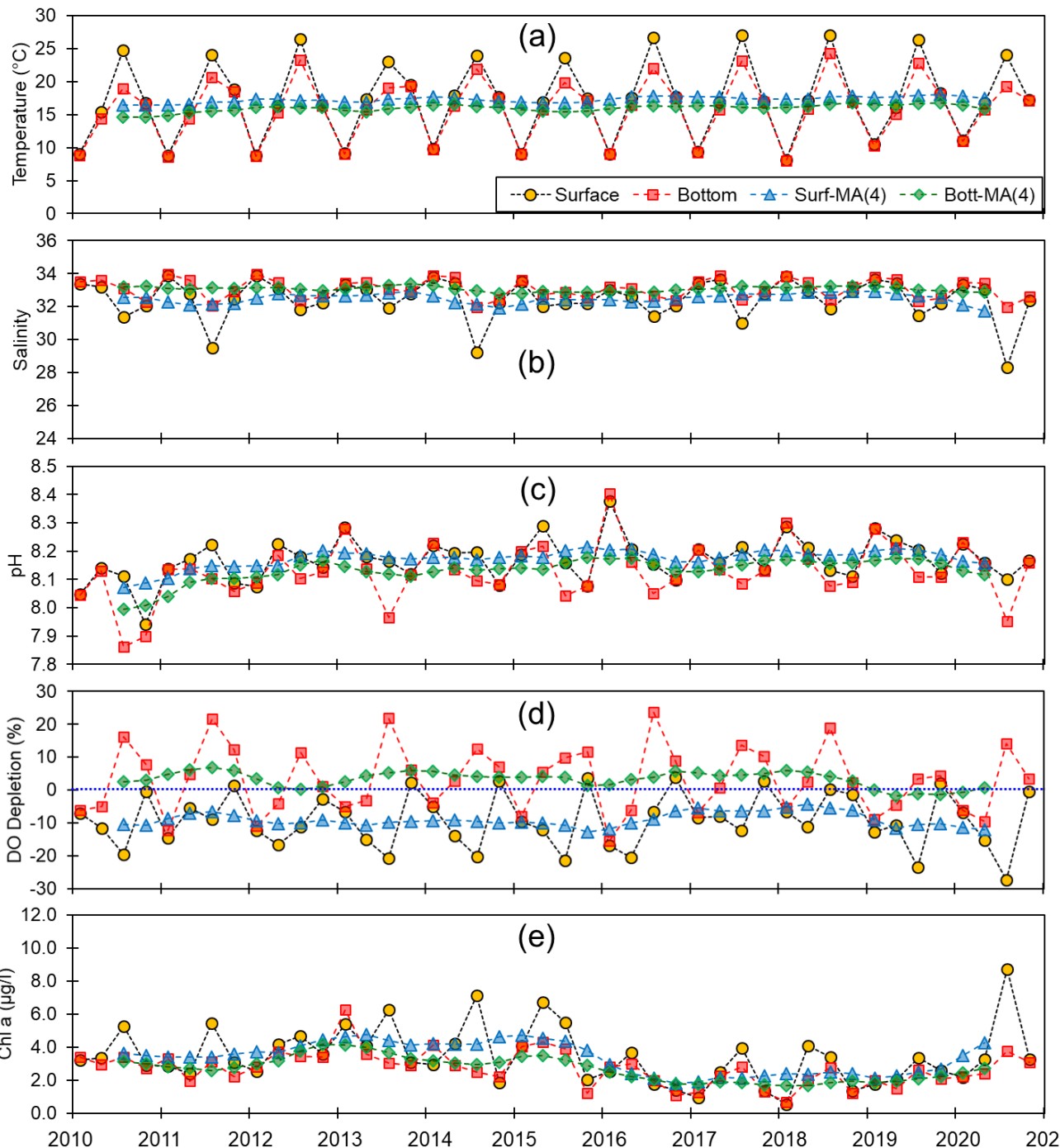

Figure 3.

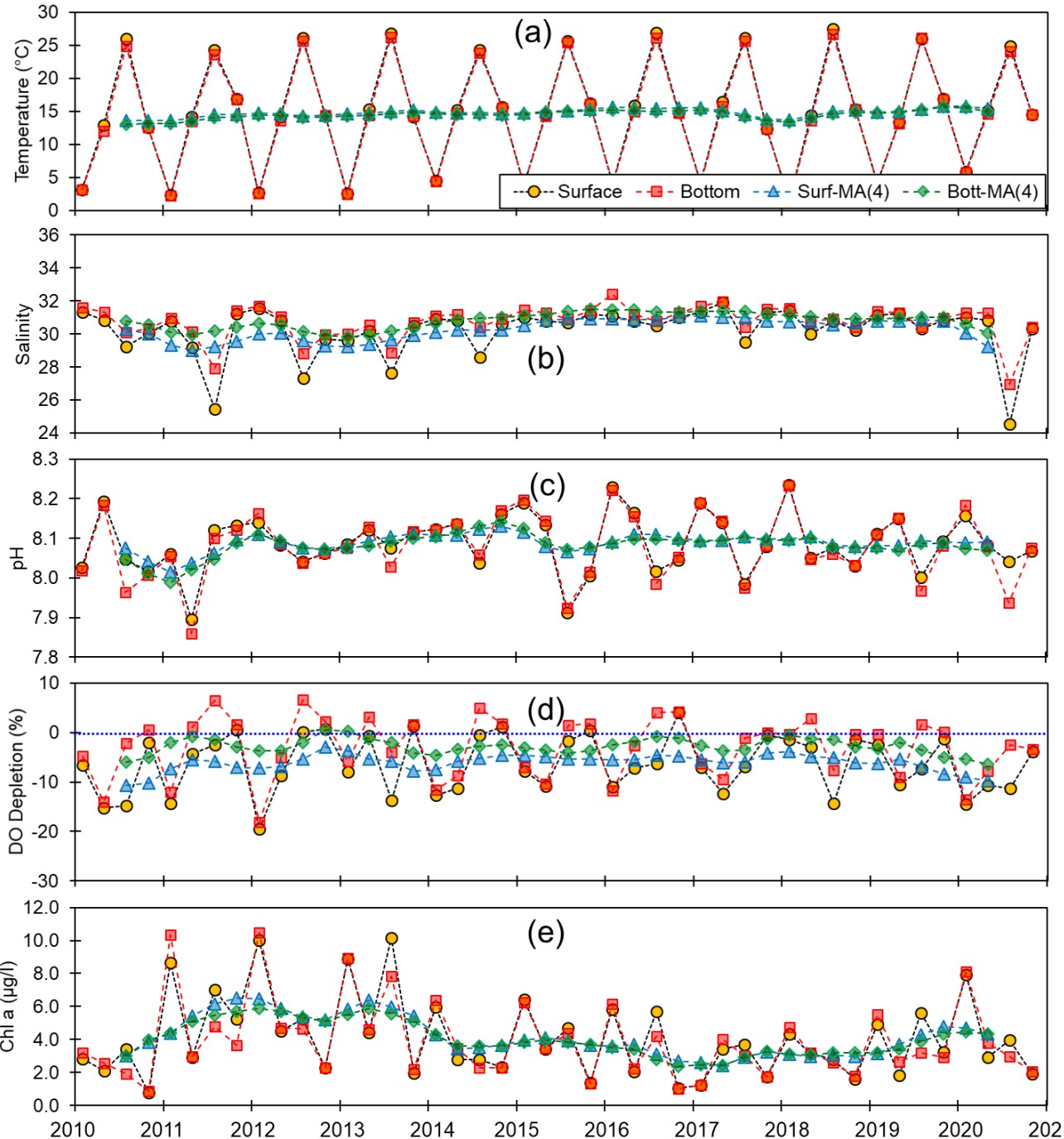

Figure 4.

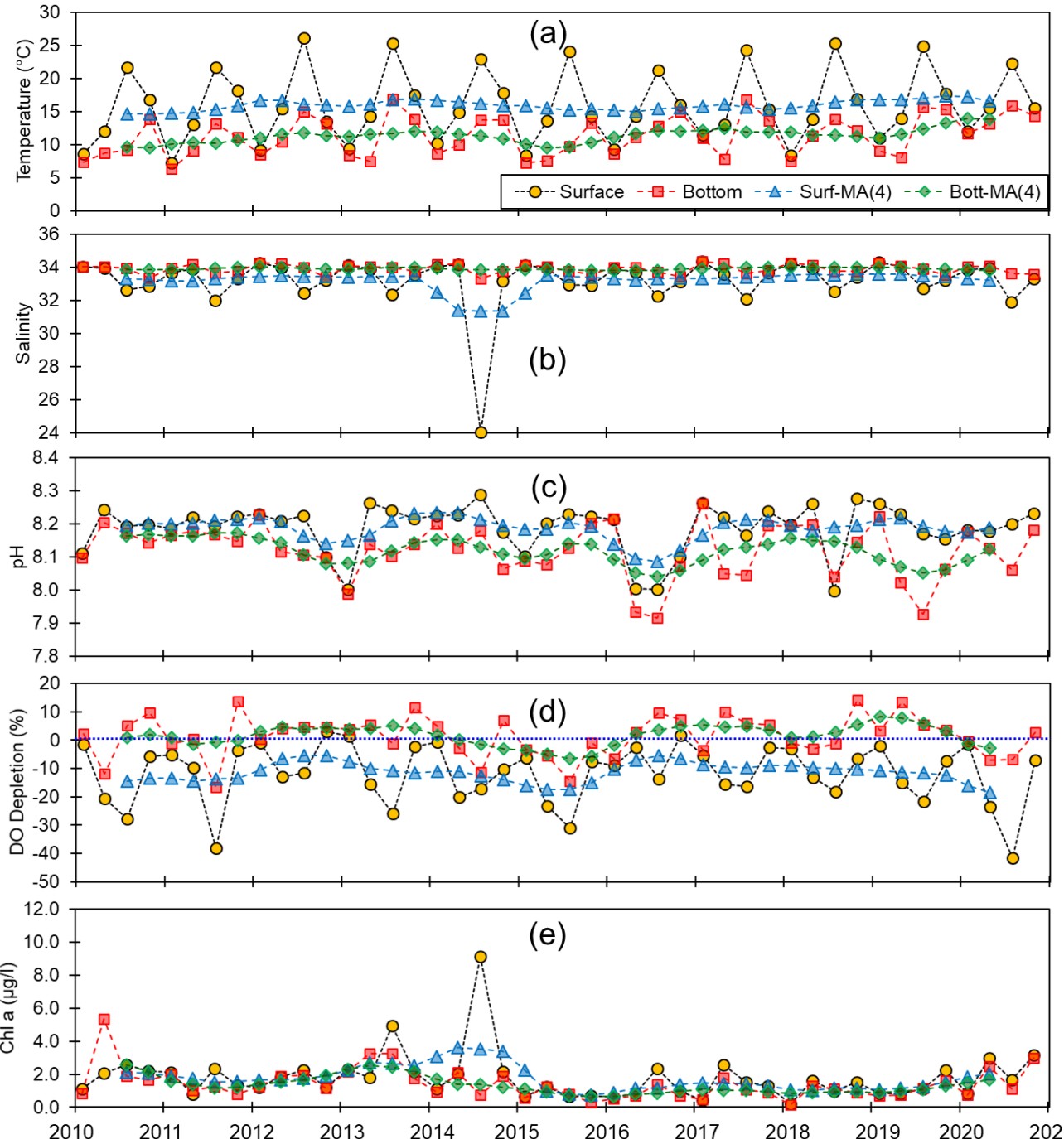

Figure 5.


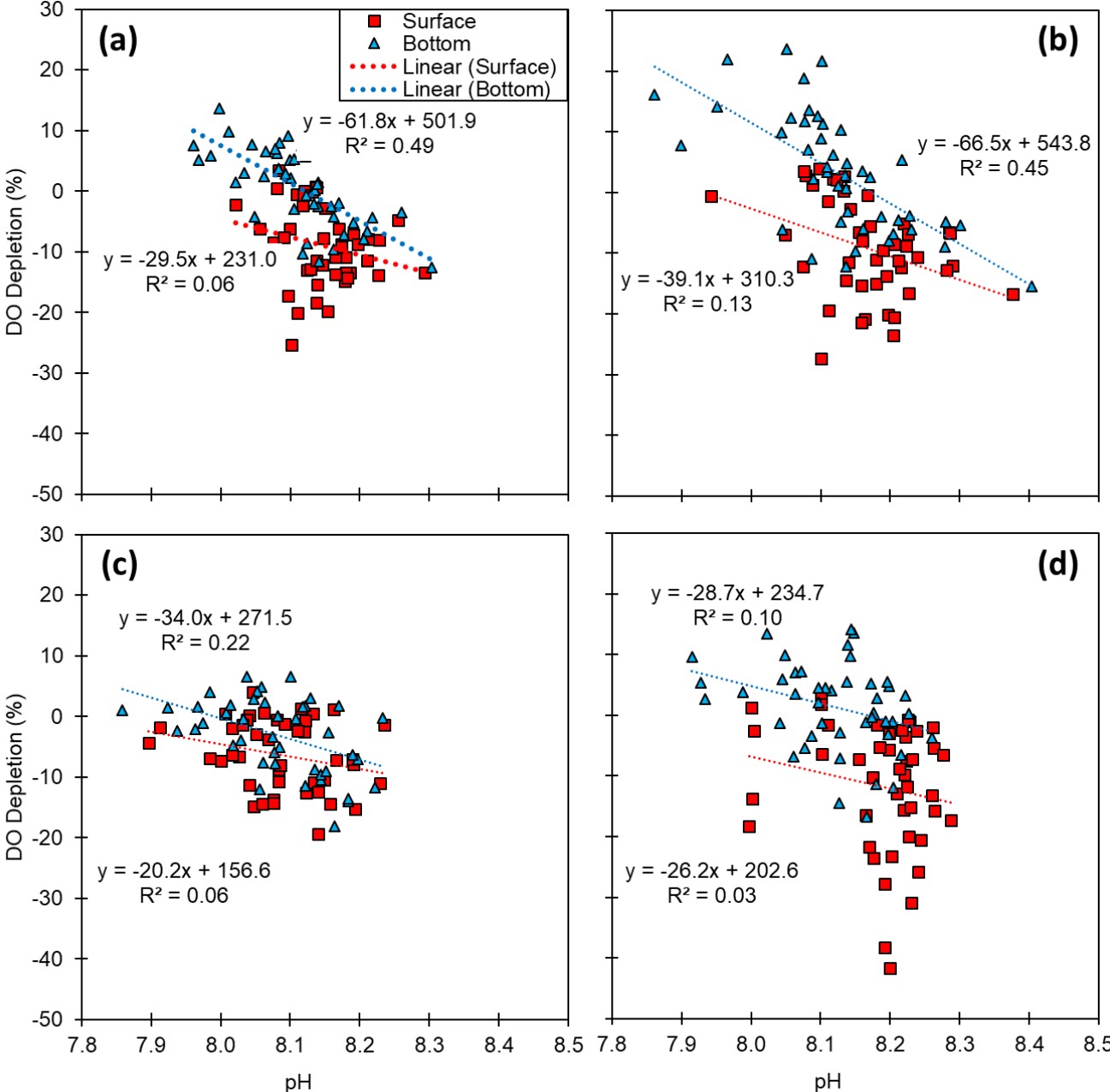

Figure 6.


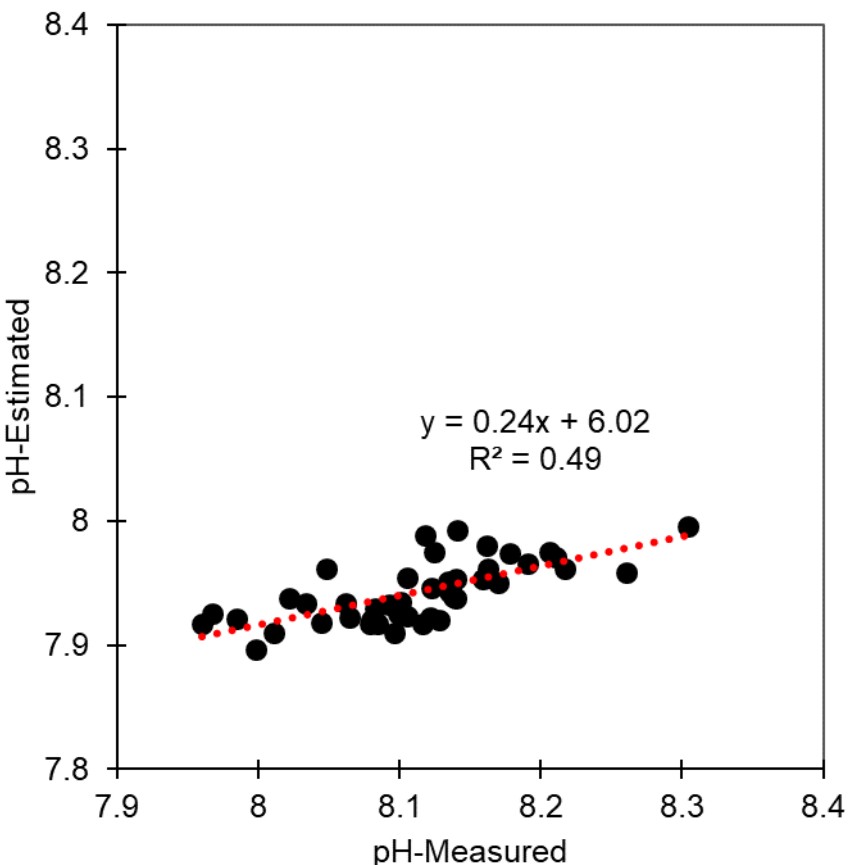

Figure 7.




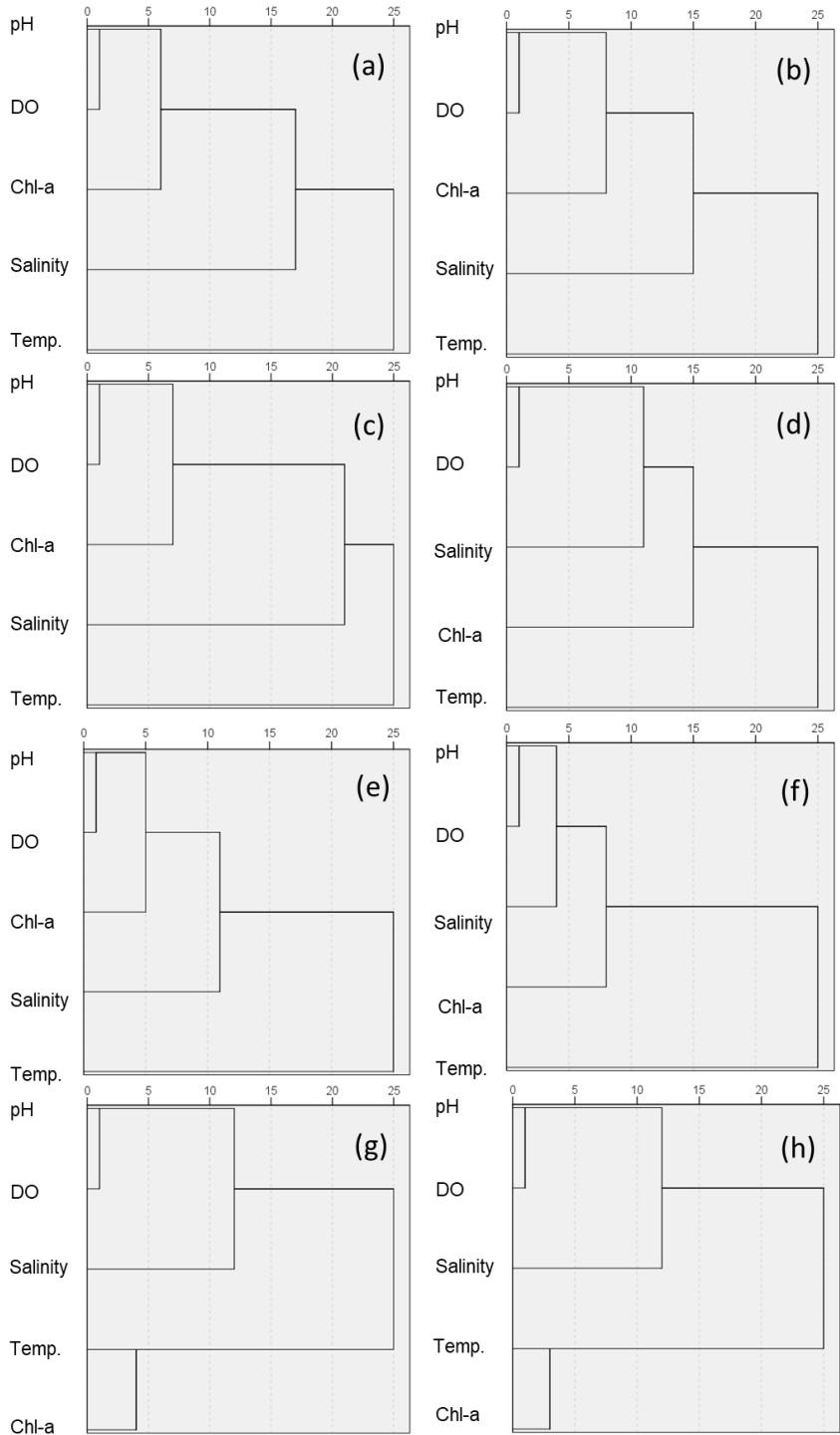

Figure 8.

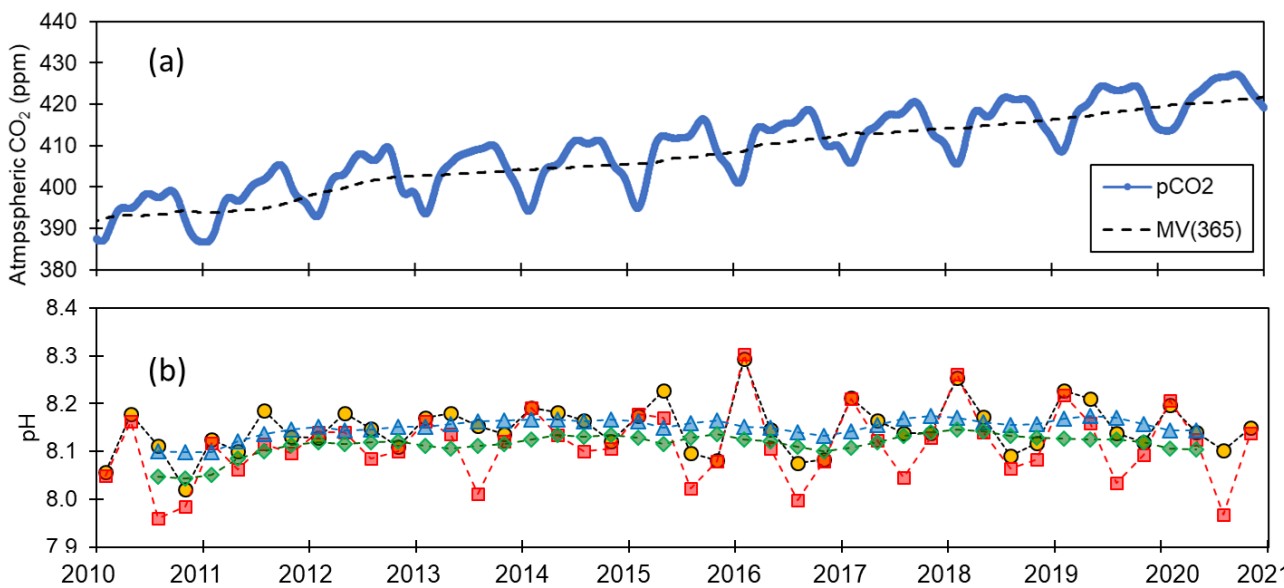

Figure 9.