# Peer review of "Long-term variations in pH in coastal waters along the Korean Peninsula"

_EGUsphere, 2024_

## Author Comment (AC1)

Response to Review #1

We would like to thank Reviewer #1 for their thorough and insightful comments, which have significantly helped improve the clarity and accuracy of our manuscript. Below, we provide detailed responses to each of the points raised:

General comments:

This manuscript investigates trends in pH over an 11-year period (2010-2020) in Korean coastal waters and claims to identify the principal drivers. The authors conclude that local biogeochemical processes are more important than ocean acidification driven by increasing atmospheric CO2, confirming what has been reported in several studies already – some of those referenced in the manuscript. Thus, I do not find any novel contributions in the manuscript, except that it presents data from a less well-studied area.

⇨ Many studies, including those cited in our manuscript, have been conducted in a limited number of offshore sites, often at considerable distances from the shore. In contrast, our study focuses on regions in close to the shoreline, primarily within 5 km, where the majority of sea-based economic activities occur. Consequently, our study provides a critical baseline for pH and other parameters, as these nearshore coastal waters have not been previously investigated.

The authors never try to further investigate which biogeochemical processes are most important. They conclude that "biogeochemical factors such as nutrient levels, biological production, oxygen conditions, and contaminant inputs likely play significant roles", although the causal linkages are not explained.

⇨ We conducted a principal component analysis (PCA), though the results are not presented here. However, the Kaiser-Meyer-Olkin (KMO) Measure of Sampling Adequacy was less than 0.5 for all datasets, indicating that PCA was not appropriate for this analysis. Consequently, we investigated the linear relationships between pH and each ancillary parameter, providing a brief explanation of the results. Contrary to findings in other regions, such as those reviewed by Zang et al. (2011), we did not observe any statistically significant

relationships among these variables. Additionally, we estimated pH based on oxygen consumption and the associated introduction of CO2, comparing these estimates with observed pH values. The strong correlation found between estimated and observed pH suggests that oxygen conditions and the associated CO2 levels play a critical role in regulating pH in Korean coastal waters. Further statistical analyses, including cluster analysis, will be incorporated in the revised version of the manuscript

In fact, I am not convinced about the contaminant inputs, which is only supported by stating that "greater contaminant inputs in Niigata, a megacity on the Japanese coast, may contribute to the observed differences" in primary production between the Korean and Japanese side of the East Japan Sea. This assertion is completely unsupported and most likely wrong, as the authors have not considered other explanations such as effects of upwelling.

⇨ The contaminant inputs in this context include nutrients and the conventional pollutants. In particular, coastal regions near megacities, where both the number and concentration of contaminants can be substantial. It is well-documented that such contaminants contribute to eutrophication, thereby enhancing primary production. The referenced study (i.e., Ishizu et al. 2019) suggests a potential increase in primary production as a result of these contaminant inputs and thus pH variation.

Not being an expert in the study area, I do think that the strong north-going current could drive upwelling along the Korean coast by Ekman transport. Dominant wind patterns could also drive upwelling along the Korean coast. Why haven't the authors investigated the hydrography of the area in more detail before making such assertion?

⇨ Upwelling occurs along the eastern coast of Korea, but it is confined to relatively small areas (e.g., off Pohang). Moreover, our sampling sites were located very close to the shore, where the influence of upwelled water is likely minimal compared to local physical processes.

Going back to the aim of identifying the drivers of pH trends, I do not think the authors have managed to do so. One of the final comments is that "our understanding of pH variability and ocean acidification in the carbonate system will remain incomplete until

we fully characterize the CO2 systems in Korean coastal oceans". This conclusion is somewhat disappointing, since the processes of the carbonate system are well described and it should be possible to tease out the importance of different processes by solving the equations in CO2SYS or similar program. The authors should be able to conduct a more thorough and detailed analysis of their data.

⇨ We estimated pH using the CO2SYS program, incorporating salinity, nutrients (dissolved silicate), AOU-derived CO2, and total alkalinity (Lines 241-244). A strong correlation was observed between the estimated and measured pH values, indicating that dissolved oxygen conditions and associated CO2 inputs significantly influence on the pH variation in coastal waters. However, it is important to note that the alkalinity values used in the CO2SYS calculations were not determined in this study but were obtained from previous research. This prior research primarily focused on the carbonate system in relatively open ocean waters, rather than in the shallow nearshore waters along the Korean peninsula. Therefore, we emphasize that to fully understand pH variations, the CO2 system in coastal waters needs to be more precisely constrained.

This brings me to my next general comment about the separation of data into three regions. If local conditions, particularly inputs of nutrients and organic matter from land, are more important then why pool sites together that have very different connectivity to land into three study regions? It would be more meaningful to separate the sites into different coastal types such as river-dominated estuaries, estuaries, lagoons, embayments, coast, etc. and perhaps use stratification patterns as another descriptor for separating sites into more similar groups. Given that the coastal systems included within each group are so diverse, averaging across these are likely to produce 'no information'.

⇨ Thank you for the suggestion. The revised version of the manuscript will include pooled data categorized by different coastal types. However, the primary objective of this study is to establish a baseline for pH in each region, which has not been thoroughly studied, and to investigate how pH has varied over the past 11 years at different sites. Our focus is on discussing pH variation at a regional scale and then integrating the data to represent the broader Korean coast. We believe that averaging across regions does provide valuable information, as this approach has been commonly used in earlier monitoring studies.

Following that, it appears that there is a lack of consistency in the sampling among cruises. Fig. 1 shows 356 sites that have been monitored, but looking at the figures in the SI it does not appear that all these sites have been monitored in each cruise. This would imply ~120 observations for each time point in each region. The consistency of monitoring is not described in the M&M, and in case that all stations are not monitored at each monitoring time point, averaging across different sites for each time point creates a bias in the time series. The authors need to address this issue more clearly.

⇨ Supplementary Figures 1-6 illustrate the surface and bottom depths across the southern, western, and eastern coastal regions, respectively, while Figure 1 provides an overview of all the sample collection sites. For reference, the map can be accessed at https://www.meis.go.kr/map/oemsBaseMap.do.

In fact, there is no description of how data were processed, i.e. how were trends assessed (with or without seasonal adjustment), how was the pH in Fig. 7 estimated and what were the assumptions, the calculation of AOU, etc.? How were surface and bottom waters distinguished and was the water column always stratified? Data processing methods seem to be introduced along the way in the results and discussion section without clear explanation as why they were chosen and what was the hypothesis. This needs to be better structured.

⇨ We plan to restructure the Methods and Materials section in the revised version of the manuscript. However, the supplementary figures displayed all the raw data, and the averaged and moving average (with seasonal adjustment) were shown in figures in the main text. Also, the estimation of pH and AOU is already described in Lines 241-244. In this monitoring study, samples were collected only from the surface and near the bottom (~1 m above the seabed). Additionally, since no vertical profiles were obtained, stratification is inferred solely from the differences between the surface and bottom layers.

Moreover, I am surprised to see that there is no apparent connection between low DO and low pH in Figure S2, S4 and S6. There is an intricate relationship between DO and pH, since oxygen consumption (mainly) produces $CO_2$. Many months have low DO without pH dropping at the same time. This is indeed surprising and makes me think that there could be issues with measuring pH (calibration?). Normally, low DO (<2-3 mg

L-1) should be accompanied by pH less than ~7.5, but this does not seem to be the case. I recommend the authors to plot measurements of DO versus pH and check for consistency in the relationships across time and region. Similarly high pH (>8.5) without extremely high chla are also unusual, confirming my assertion that something could be wrong with the data.

⇨ The pH sensor was calibrated daily before sample measurements using three standard buffer solutions (pH 4, 7, and 10), following the manufacturer's guidelines. Dissolved oxygen (DO) concentrations do not always accurately reflect the degree of oxygen consumption or $CO_2$ production. Therefore, we used DO depletion (%) as a more appropriate measure instead of DO concentration. Graphs of DO concentrations and pH will be included in the supplementary materials.

⇨ In relatively stratified deep systems, low DO levels, indicative of oxygen consumption, typically correlate linearly with a decrease in pH. However, in shallow coastal waters where wind, currents, and tidal forces are active, $CO_2$ may escape from the water column before it can fully dissolve. As a result, not all oxygen consumption leads to a pH drop. Consequently, in many water bodies, pH does not always correlate with DO consumption (e.g., Li et al., 2024). Additionally, high DO levels do not necessarily indicate high primary production (e.g., chlorophyll a concentration) but can result from episodic wind events (e.g., Pidgeon and Winant, 2005) or production by macroalgae (e.g., Kelp, Frieder et al., 2012). Thus, pH does not always strongly correlate with chlorophyll a.

Finally, the language is not always clear and it is difficult to see whether it is due to language difficulties or lack of understanding of the biogeochemistry. I found statements that were wrong, unclear and misleading, but the cause of this is not clear to me.

⇨ We apologize for any difficulties in reading this manuscript. However, the language has been reviewed by a professional manuscript-language service (e.g., Elsevier English Language Service) and a commercially available scientific manuscript review system.

Specific comments:

67: High alkalinity would actually buffer against acidification, so this can hardly be an explanation to higher trends in the Mediterranean Sea!

⇨ We will clarify this sentence. However, we did not imply that high alkalinity decreases pH. The sentence aims to describe the unique biogeochemical and hydrodynamic characteristics of some places in the Mediterranean Sea.

86: Pristine implies a state before human disturbance. I doubt that the east coast is entirely undisturbed, so better to say "relatively undisturbed".

⇨ The term "pristine" has been revised. However, "pristine" is commonly used in the literature to describe relatively "clean" conditions, and numerous articles have employed this term.

145: The authors claim that OA is the foremost concern for coastal ecosystems. This claim is unsupported and in my view not correct. Eutrophication, food-web alteration from overfishing, physical disturbance from bottom trawling are pressures that are prominent in the coastal zone and have larger impact on organisms and habitats. This sentence should be deleted.

⇨ Thanks for your opinion. In the context of global warming (associated with $CO_2$) and its impact on water quality and biological productivity (particularly primary production), ocean acidification represents a significant threat to coastal ecosystems. Given the rapid decline in pH levels in many areas, ocean acidification is of particular concern for these ecosystems.

150: Differences between surface and bottom water pH is driven by production in the surface increasing pH and respiration in the bottom decreasing pH. This is not very clearly expressed here and it needs to be.

⇨ We will clarify this by rearranging the information. At the surface, primary production releases oxygen, while oxygen consumption occurs in the bottom water. These oxygen conditions are directly linked to $CO_2$ dynamics, leading to pH variations. The detailed explanation of this process can be found in the section "3.3 Factors Influencing pH Variations."

152: Was the trend exactly the same for surface and bottom? That seems odd.

⇨ The moving average (with 4 cycles and seasonal adjustment) revealed a discernible increasing trend in both surface and bottom depths (Figure 2). The slopes of the surface and bottom datasets differed only at the 6th decimal place, which has been rounded, making them appear identical.

153: F=00003 and not significant, but the slope was 0.0009±0.0001 which with an ordinary t-test gives a highly significant slope. This does not match and needs to be checked.

⇨ Thank you for highlighting this issue. We provided incorrect numbers, and they should be 0.000029 yr$^{-1}$ (±0.000007) and 0.000035 yr$^{-1}$ (±0.000009) for the surface and bottom, respectively. However, these corrections do not affect our overall conclusion. Nonetheless, we misinterpreted the statistic. The F-value in ANOVA regression assesses the overall significance of the regression model, calculated as the ratio of the variation between sample means to the variation within the samples. An F-value of 0.0003, which is lower than the significance level α = 0.01, indicates statistical significance for the regression model. However, the R-square value for the linear regression was 0.29 (n=40) for both surface and bottom depths, indicating that the trend is not statistically significant or meaningful. Additionally, as you pointed out, a change of 0.00003 per year may not be substantial.

166-167: Coastal eutrophication stimulates production in the surface layer causing pH to rise and respiration in the bottom layer causing pH to decrease. The sentence does not articulate this very well and it is misleading or actually wrong.

⇨ We will clarify it. The Pacific side is influenced by the Kuroshio Current, whereas the East Sea side is affected by coastal eutrophication. These factors contribute to the exacerbation of pH differences between the two regions (see, Ishizu et al., 2019).

180: Can this be elaborated more specifically? What biogeochemical properties are driving pH dynamics?

⇨ We will clarify and elaborate it. The biogeochemical properties that differ between open and coastal oceans include biological productivity, chemical inputs from rivers and sediments, and wind stress, among other factors. These variations all

contribute to differences in pH between open and coastal oceans.

198: Elaborate how this will influence pH?

⇨ We will clarify it. However, the relationship between low O2 (hypoxia), CO2, and pH is well-established and was also discussed in the Introduction.

207: "long-term pH trends may exhibit a negative correlation with temperature". The authors refer to CO2 solubility, but warmer water can contain less CO2 in equilibrium! This should cause a positive correlation. However, there is another factor that the authors have overlooked, the issue of shifting equilibrium constants in the carbonate system. As the water warms, the carbonate system shifts towards more dissolution, producing more H+ ions and thereby lowering pH. This might be more relevant than the equilibrium with the atmosphere due to the slower kinetics of the gas exchange.

⇨ Thank you for your suggestions. We will expand on this topic. Thermodynamically, an increase in temperature leads to an increase in the dissociation constant, which in turn results in a decrease in pH. While temperature can influence gas solubility and degassing, potentially creating a positive correlation between pH and temperature, its more substantial impact occurs through various biogeochemical processes. Elevated temperatures can enhance water column stratification and contribute to bottom water hypoxia, leading to CO2 accumulation and redox reactions that lower pH. Additionally, higher temperatures can reduce primary production by impairing algal metabolic activity, resulting in decreased CO2 consumption in the water column. Overall, these factors suggest that temperature has a negative impact on pH.

242-246: This should be described under the Materials and methods section. The calculations are also based on the assumption that DO consumption results from respiration and nitrification?

⇨ It will move to Materials and methods section. The AOU calculation accounts respiration, since Redfield ratio accounts oxygen consumption through organic matter diagenesis.

248: "pH in Korean coastal waters is primarily controlled by oxygen conditions". This is not correct. Both DO and pH result from an imbalance between respiration and

production.

⇨ We will restate and clarify this point. Apparently, only AOU showed a correlation with pH, despite some offset between estimated and measured pH values. This indicates that oxygen conditions and associated CO2 dynamics play a crucial role in pH dynamics.

254: Why would atmospheric CO2 have increased more along the Korean coast? This does not make sense. The atmospheric concentration is relatively constant across the northern hemisphere as well as the southern hemisphere.

⇨ East Asia has shown that the atmospheric CO2 has been increased more rapidly than other continents (see, Yeh et al., 2023).

256: Is this because different periods are compared? The authors cannot compare trends based on different periods.

⇨ We are looking the yearly increasing rates, not single month (see, https://gml.noaa.gov and https://www.index.go.kr/unity/potal/main/EachDtlPageDetail.do?idx_cd=1399, for Hawaii and Anmyendo, respectively).

295: Same comment as for L. 248.

⇨ See above

Technical comments:

37-38: "though land-based input can be significant in river-dominated estuaries".

⇨ Corrected

42-43: "Hawaii Ocean Time-series (HOT)"

⇨ Corrected.

43-44: "European Station for Time-series in the Ocean of the Canary Islands (ESTOC)"

⇨ Corrected.

52: delete "a" before "long-term"

⇨ Corrected.

53: "surface water pCO2"

⇨ Corrected.

56: replace "plankton" with "coccolithophores" to be more specific.

⇨ Plankton includes also pteropods and various other organisms. Therefore, the term "plankton" is more appropriate in this context.

59: emphasize "affecting many trophic levels" – not all.

⇨ Corrected.

63: insert "atmospheric" at beginning of line.

⇨ Corrected.

70: "decreased only by"

⇨ Corrected.

74: "due to increasing atmospheric CO2"

⇨ Corrected.

81: "possibly related to decreasing nutrient input (Provoost et al., 2010)"

⇨ Corrected.

91-92: "impacts of atmospheric CO2, warming and changing environmental conditions"

⇨ Corrected.

95: Should be "Korean Marine Environment Management Corporation (KOEM)"?

⇨ "Korea" is correct.

96: replace "assessing" with "monitoring"

⇨ Corrected.

102: delete "a comprehensive 11-year dataset to assess" – it is repetitive

⇨ Corrected.

104: delete "of" before "< 10 m"

⇨ Corrected.

104: Should be "The sites", not "These sites"

⇨ Corrected.

110: "Large river systems are found in the west and south"

⇨ Corrected.

115: "Korean Coastal Current receiving inputs from the Kuroshio warm current"

⇨ Corrected.

119: replace "provide" with "deliver"

⇨ Corrected.

L.131: "KOEM research vessel" and "were measured with a CTD"

⇨ Corrected.

141-142: "the chl a was extracted from the filters and measured with a fluorometer"

⇨ Corrected.

146: start with "OA is of particular concern along the"

⇨ Corrected.

149: "Monthly averages for February, May, "

⇨ Corrected.

154: "at least over the study period. The observed pH trend"

⇨ Corrected.

161: "driven by rising atmospheric CO2 and variations in local salinity and primary production"

⇨ Corrected.

172: delete last part of sentence. Does not convey anything.

⇨ Corrected.

178: "regional trends in pH fluctuations can differ substantially from OA."

⇨ We think the current version stated more clearly.

211: A temperature increase of 0.0009 yr-1 is nothing. This cannot be a clear increasing trend, i.e. significant?

⇨ Corrected.

214: Freshwater in Korea might have a lower alkalinity in freshwater input than the ocean end-member due to the geology, but this is not the case everywhere. Watersheds with limestone typically have higher alkalinity than the ocean end-member.

⇨ Elaborated. The bedrock of watersheds in Korea is not limestone.

218: This lack of relationship is most likely due to comparing averages, which removes most of the variability between pH and salinity. Try look at the raw data!

⇨ Row data values are provided in the supplementary information. These values did not show any significant correlations.

221: Replace "increased" with "elevated".

⇨ Corrected.

230: "Reduced DO concentration primarily results from biological respiration …."

⇨ Corrected.

232: DO can degas from the surface water, if and only if DO is supersaturated.

⇨ This sentence indicates that DO solubility decreases under warming conditions. Additionally, when DO is supersaturated, it can diffuse into the atmosphere.

238: "net DO consumption"

⇨ Corrected.

240: Actually, the correct term to use for this measure is Apparent Oxygen Utilisation (AOU). Please use this consistently.

⇨ Corrected.

245: "range of the observed"

⇨ Corrected.

---

## Author Comment (AC2)

Response to Reviewer #2.

We would like to thank Reviewer #2 for their thorough and insightful comments, which have significantly helped improve the clarity and accuracy of our manuscript. Below, we provide detailed responses to each of the points raised:

Dear Authors,

This study entitled "Long-term variations of pH in coastal waters along the Korean Peninsula" uses data (including temperature, salinity, pH, and DO) collected by a government agency in Korea from 2010-2020. Long-term data is rare. The authors suggest that coastal biological effects are more important than increasing atmospheric CO2. The data quality is not described in this study. The application of linear regression can be improved. The discussion on biogeochemical processes involving pH should be reevaluated. The major comments are as follows.

➔ Our study concentrated on regions within a 5 km proximity to the shore, where the majority of sea-based economic activities take place. This investigation establishes a baseline for pH and other key parameters, as these nearshore coastal waters had not been thoroughly examined in prior research. The methodologies for data acquisition, sample collection, and parameter calibration are comprehensively detailed in the Methods and Materials section. In the revised version, we have included the results of the statistical analysis.

1. Though the pH probe itself can reach a higher resolution, the resolution of these three pH buffers is unclear. Therefore, the uncertainty for this long-term trend of pH is unclear. Data QA and QC are unclear. What is the standard deviation for these average numbers? Moreover, oceanographers usually use the spectrophotometric method to measure pH or pH calculated by total alkalinity and dissolved inorganic carbon to study long-term pH changes. The pH value measured by a probe can be affected by its salinity. As the salinity varied in the surface water, the effect of salinity changes on this probed pH may also involve the pH changes.

If we understand the comments correctly:

⇨ The revised version will include a table describing the basic features of the data.

We have provided the entire raw and averaged data sets in Supplementary Figures 1-6, which display deviations from the average. The pH probe was calibrated at three points using buffers with pH values of 4, 7, and 10, all of which exhibit acceptable variation with temperature. Calibration was performed daily, and the pH probe has an accuracy of ±0.002. Based on these factors, our pH data set is suitable for scientific analysis. While the spectrophotometric method offers better accuracy and precision, it is not cost-effective. Portable pH measurement devices, as used in long-term studies (e.g., Ishizu et al., 2019), provide a practical alternative. Salinity can serve as a proxy for alkalinity (with lower salinity indicating lower alkalinity) and thus may relate to pH variation. However, salinity itself does not directly affect pH measurement.

2. The analysis method. There are already many new methods that can analyze multiple parameters. The authors only use linear regression. The authors can try to use a better method that can systematically analyze several variables at the same time, such as principle component analysis or similar statistic methods.

➔ Thank you for the suggestions. The revised version will include an analysis of statistical techniques. Principle component analysis (PCA) was conducted in this study. However, the values of Kaiser-Meyer-Olkin Measure of Sampling Adequacy were < 0.5 for all depths and locations, suggesting that the PCA application is not adequate.

3. The definition of pH should be listed here as this study tries to describe the change in pH. Furthermore, why should pH be linearly correlated with other parameters? How and why is pH correlated to DO? Can the authors list the chemical equations to show that they are linear?

➔ pH measures the concentration of $[H^+]$, which can be influenced both directly and indirectly. Directly, pH is affected by $CO_2$ introduction (assuming no buffering effects) through the chemical reactions (see, Introduction). Indirectly, pH can be influenced by various environmental factors such as temperature, salinity, and primary production (see, Dickson, 2010). Low DO levels indicate oxygen consumption, which leads to $CO_2$ production. This increase in $CO_2$ decreases pH.

4. The effect of mixing between freshwater and sea (salinity gradient) on pH variation is

non-linear.

⇨ In general, the influence of salinity on pH (or alkalinity) is limited under conditions where salinity is less than 20 (Carstensen and Duarte, 2019). Most of our study sites exhibited salinity levels greater than 23, indicating sufficient alkalinity to buffer pH changes.

5. The authors separated their data into the surface and bottom water in this study. However, the authors did not separate their discussion. In Cai et al. (2011), synergistic acidification is for bottom water. Surface water in the coastal region has been known as a high-productivity region with a high pH value. Is it possible that, though the authors used a long-term dataset, the resolution of the sensor and their standard variation, as well as the analysis method, is not sensitive enough to quantify the effect of acidification?

➔ We discussed surface and bottom waters separately where relevant, but we also addressed both simultaneously when applicable. For example, temperature did not correlate well with pH in either surface or bottom waters. However, when discussing DO, we focused on bottom waters, following the approach of Cai et al. (2011). If measurements are consistent in terms of location and technique over an extended period, they can reveal trends. We investigated long-term pH trends in coastal waters because the annual variation in pH is minimal (e.g., -0.0019 per year over 20 years as observed in HOT).

6. The grammar should be thoroughly checked.

⇨ Thanks for the comments.

---

## Author Response (AR2)

Reviewer #1

Suggestions for revision or reasons for rejection

This study has improved. The authors organize and analyze the water temperature, salinity, DO, pH, and Chl-a data over Korea's coast. They suggest that primary production is the key to affecting pH, while other biogeochemical processes can also affect pH variations.

⇨ We sincerely appreciate your time and effort in reviewing our manuscript. Following the incorporation of the first-round reviewers' suggestions and corrections, the manuscript has undergone significant improvement beyond our initial expectations.

⇨ This study aims to establish a baseline for pH changes along the Korean coastal waters. The key finding is that there was no significant variation in pH, or in the other environmental parameters measured (except for temperature), throughout the study period. This suggests that pH and the other parameters in Korean coastal waters were not primarily influenced by atmospheric CO2 increase (or global climate crisis), at least, during the study period, but were likely affected by local factors, such as primary production and associated dissolved oxygen dynamics. However, with the accelerating pace of global warming, it is anticipated that pH in Korean coastal waters will be more significantly impacted by climate change in the near future.

Major comments:

"DO depletion %" is rarely used in marine or aquatic studies. Do the authors mean DO saturation? Line 298-Line 300 is the equation or not? I am not sure. How is this subtracting concentration converted to percentage? Equations are needed to clarify this.

⇨ We have added an equation to describe DO depletion (or AOU) in Method and Material section as well as in the caption of Fig 6 for DO depletion (%), which represents the proportion of the difference between the dissolved oxygen (DO) concentration at saturation and the in situ DO concentration, relative to the saturation concentration. Since DO saturation is a function of temperature and

salinity, saturation concentrations vary on daily and seasonal timescales. Thus, relying solely on DO concentration comparisons may not accurately reflect actual oxygen consumption. In this context, DO depletion (or consumption) provides a more precise measure of oxygen consumption driven by biological activity (e.g., respiration), which is directly associated with $CO_2$ release and pH variations. Furthermore, expressing DO consumption (or depletion) as a concentration (e.g., in molarity units) may not adequately reflect the severity or extent of oxygen depletion. For instance, during summer, when high temperatures and salinity reduce DO solubility, a lower DO depletion concentration may still indicate significant oxygen consumption. Additionally, we observed that the term DO depletion (either as a percentage or concentration) is used under various names, such as apparent oxygen utilization (AOU). Therefore, we believe that using the term "DO depletion (%)" in this manuscript is the most appropriate way to describe our dataset and its interpretation.

Line 168-170: While the local processes may vary, can authors estimate if this study underestimates or overestimates their pH results by following this assumption?

⇨ We clarify the senstence. Biological activities, such as respiration, release $CO_2$, which leads to an increase in [H+] and consequently a decrease in pH. However, in shallow coastal environments, not all of the released $CO_2$ remains in the water column; a portion may escape into the atmosphere through air-sea gas exchange via diffusion, wind, and wave action. As a result, our assumption may overestimate the extent of pH decrease.

Minor comments

Line 326, "DO" alone or "biogenic DO changes" can affect pH?

⇨ In the context of cluster analysis, DO alone may be more suitable, as variations in DO result not only from biological production but also from changes in temperature and salinity.

L332-334 These words should be modified as the authors have cited several references to indicate that this pH reduction is nearly 0.0016 per year.

⇨ Thank you for your suggestion. The sentence has been revised to: "...which results in a pH decrease of approximately 0.002 per year (Solomon et al., 2007)."

Line 369-373: A few references are needed to support these two sentences.

⇨ We added several references that were already cited in this manuscript:
Kroeker et al. (2013), Impacts of ocean acidification on marine organisms: quantifying sensitivities and interaction with warming. Global Change Biology, 19(6), 11881-1896.
Lowe, A. T., Bos, J., & Ruesink, J. (2019). Ecosystem metabolism drives pH variability and modulates long-term ocean acidification in the Northeast Pacific coastal ocean. Scientific Reports, 9(1), 963.
Breitburg, D., Levin, L. A., Oschlies, A., Grégoire, M., Chavez, F. P., Conley, D. J., ... & Zhang, J. (2018). Declining oxygen in the global ocean and coastal waters. Science, 359(6371), eaam7240.

⇨ They are discussing about the impact of pH on fish and shellfish (Kroeker et al., 2013; Lowe et al., 2019) and global ocean warming on pH variation (Breitburg et al., 2018).

Line 450, 493, lowercase for 2 and 3

⇨ Thanks for pointing this out. Corrected.

Figure 7. Use dark open circles as the markers.

⇨ Corrected as suggested.

Figure 5. The lowest label on panel B on the Y-axis is missing.

⇨ Thanks for pointing this out. Corrected.

The overall quality of these figures should be improved.

⇨ We have thoroughly reviewed the figure and made several adjustments.

Figure titles are missing except for Figure 1, which lacks real figure captions.

⇨ We included figure titles.